# Divergent photochemical ring-replacement of isoxazoles

Yan Xu[1,2], Lorenzo Poletti[1], Enrique M. Arpa [1], Baptiste Roure [3] ✉, Alessandro Ruffoni [4] ✉ & Daniele Leonori [1] ✉

Isoxazoles, oxazoles, and other five-membered heteroaromatics are prevalent motifs in core structure of pharmaceuticals and agrochemicals. In early-stage drug discovery, it is common practice to prepare libraries of analogues featuring different heterocyclic cores and this generally requires a de novo synthesis for each scaffold. A valuable but currently unavailable strategy would involve the possibility for direct heterocycle "ring-replacement". Here we report a photochemical platform for the selective conversion of isoxazoles into oxazoles, pyrazoles, pyrroles, and isothiazoles by exploiting excited-state reactivity. Starting from a successful isoxazole-to-oxazole transformation, we uncover position-sensitive reactivity that prompted computational investigation. These insights guide a systematic reactivity survey and reveal a solvent-controlled deconstruction–reconstruction pathway via α-ketonitrile intermediates. This approach enables scaffold diversification without de novo synthesis, affording access to five distinct azole classes under mild conditions. The method's selectivity, functional group tolerance, and late-stage applicability suggest broad utility in heterocyclic library design for pharmaceutical research.

The ability to modify complex molecules at late stages, without the need for de novo synthesis, has become a central objective in modern synthetic chemistry[1]. Considering (hetero)aromatic molecules, significant advances have been made in the way we perform peripheral modifications (e.g., cross-coupling and C–H activation)[2,3]. However, direct replacement or remodeling of the core (hetero)aromatic scaffold remains a largely unsolved challenge, despite its potential to accelerate chemical space exploration in pharmaceuticals, agrochemicals, and functional materials[4,5].

Among five-membered heteroaromatics, isoxazoles and oxazoles occupy a privileged position, frequently appearing in marketed drugs and advanced materials (Fig. 1a)[6,7]. These heterocycles can serve as bioisosteres for ketone, ester, and (hetero)aryl groups, often offering benefits such as improved metabolic stability, reduced lipophilicity, and enhanced potency[8,9]. Their privileged status makes them invaluable tools in modern drug discovery and development.

A classic approach in early structure-activity relationship (SAR) studies involves the preparation of substrate libraries in which a specific heterocyclic system is replaced by closely related bioisosteres[10–12]. A representative case is the development of the isoxazole-based SETD2 inhibitor 1, where oxazole, imidazole, and pyrazole analogs were individually synthesized and evaluated (Fig. 1b)[13,14]. While effective for lead identification, such strategies typically necessitate the de novo synthesis of each variant, which becomes especially challenging when the heterocycle is embedded within the molecular core[15,16]. Consequently, a late-stage method for heterocycle "ring-replacement" would offer significant value by enabling rapid access to bioisosteric analogs from a common lead derivative[17,18].

Isoxazoles offer a compelling platform for this strategy. Early reports demonstrated their capacity to rearrange photochemically to oxazoles and α-ketonitriles, but the reactions were typically unselective, low-yielding, and limited to a narrow set of derivatives[19–30]. Due to

[1]Institute of Organic Chemistry, RWTH Aachen University, Aachen, Germany. [2]College of Chemistry and Environmental Engineering, Shenzhen University, Shenzhen, China. [3]School of Chemistry, University of Manchester, Manchester, UK. [4]Otto Diels – Institute of Organic Chemistry, Christian Albrecht Universitat zu Kiel, Kiel, Germany. ✉e-mail: baptiste.roure@rwth-aachen.de; aruffoni@oc.uni-kiel.de; daniele.leonori@rwth-aachen.de

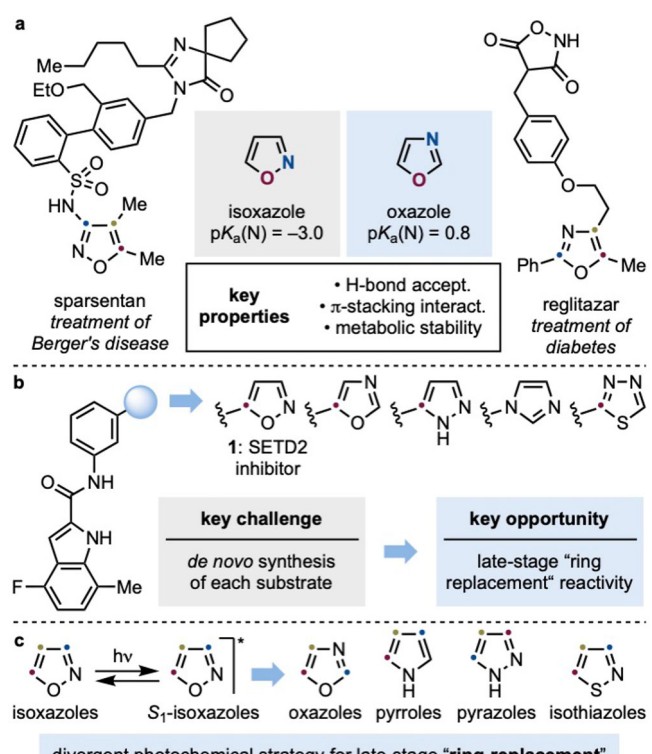

**Fig. 1 | Relevance of oxazoles and other azoles in medicinal chemistry.**
**a** Relevance of isoxazoles and oxazoles in pharmaceuticals. **b** Ring replacements in SAR studies towards the identification of a SETD2 inhibitor **1**. **c** This work uses photochemistry to convert isoxazoles into other heteroaromatics.

these challenges, the photochemistry of isoxazoles has remained unexplored, and no general framework exists to harness their reactivity in a selective or synthetically useful way.

Here we report a photochemical platform for the selective conversion of isoxazoles into other azoles, including oxazoles, pyrazoles, pyrroles, and isothiazoles (Fig. 1c). Beginning with the efficient isoxazole-to-oxazole rearrangement of 5-phenyl isoxazole (**2a**), we observed unexpectedly different behavior across closely related analogs. This prompted a mechanistic investigation using DFT, which revealed how small changes in the isoxazole substitution pattern drastically influence excited-state reactivity. Leveraging these insights, we constructed a comprehensive reactivity map encompassing 24 substituted isoxazoles and identified structural features that govern productive rearrangement. Beyond rearrangement, we discovered that solvent modulation could redirect excited-state reactivity away from C−C bond fragmentation, yielding α-ketonitrile intermediates that serve as branch points for further heterocycle formation. Using this principle, we developed a divergent deconstruction−reconstruction platform capable of generating five distinct heterocyclic families from readily available isoxazole precursors[31].

## Results and discussion

In developing a strategy for the conversion of isoxazoles into other heteroaromatics, we drew inspiration from the early photochemical experiments from Ullman and Singh[26,27]. These pioneering studies demonstrated that simple photoexcited isoxazoles can undergo a series of structural rearrangements leading to either azirine intermediates or α-ketonitriles[32]. However, pioneering studies from Pavlik et al. demonstrated its synthetic potential in isoxazole-to-oxazole isomerization, although their conditions often led to complex reaction outcomes, low chemical yields, and mixtures of isomeric

products[19–25,28–30]. More recently, Baumann has translated the isoxazole-to-oxazole conversion under photo-flow settings using high-energy Hg-lamps, but this reactivity was applied to a small subset of derivatives that do not capture the full chemical space options around the isoxazole core[33]. As a result, the synthetic scope and selectivity of isoxazole photochemistry remain essentially unknown.

We initiated our study with 5-phenyl isoxazole (**2a**) and explored its conversion into either the corresponding oxazole (**2b**) or α-ketonitrile (**2c**), depending on the reaction conditions (Fig. 2a). Under irradiation at $\lambda = 310$ nm, we observed some remarkable differences in the reaction outcome on the basis of the solvent and additive used. Specifically, irradiation in MeOH led to the selective formation of **2b** in good yield (entry 1). Interestingly, other polar protic solvents lead to considerably lower conversions and significant decomposition (entries 2 and 3). We then evaluated a few additives, and while both standard acid and base did not improve the reaction efficiency (entries 4 and 5), the use of 20 mol% of 2,6-lutidine delivered **2b** in 85% yield (entry 7)[34]. These conditions could be scaled up in a single batch condition (8 mmol) using the same irradiating apparatus to provide **2b** in 80% yield (see the Supplementary Information Section 7 for more details). The use of non-polar solvents (e.g., trifluorotoluene and EtOAc, entries 8 and 9) had a dramatic switch in the reaction, now providing nitrile **2c** as the major product. Pleasingly, the use of DCE completely suppressed the formation of **2b** and provided access to **2c** in 72% yield.

Encouraged by these results, we explored how changes to the position of the Ph substituent on the isoxazole ring might influence reactivity (Fig. 2b). In general, all isoxazoles discussed below have been subjected to six photochemical conditions (A–F), and the results reported represent the highest yielding ones. Strikingly, the reactivity of **2a** was not preserved on its constitutional isomers. Indeed, the 4-phenyl isoxazole (**3a**) did not undergo any reactivity when irradiated ($\lambda = 310$ nm) in the presence of 2,6-lutidine as the additive and MeOH as the solvent. However, partial conversion was achieved with more energetic irradiation ($\lambda = 254$ nm), yielding oxazole **3b** in modest yield (23%, conditions E). Furthermore, the 3-phenyl analog (**4a**) proved photochemically inert under all tested conditions, even under prolonged irradiation.

This sharp change in photochemical reactivity is difficult to rationalize based on the limited knowledge in the area, and therefore, we have run full reaction analysis by DFT. The proposed mechanism leading to the isoxazole-to-oxazole (**A**-to-**E**) isomerization or isoxazole-to-α-ketonitrile (**A**-to-**G**) ring-opening has generally been approached considering that irradiation populates the isoxazole π,π* singlet excited state ($S_1$-**A**) (step i), subsequently evolving to a vinylnitrene (**B**) via fast N−O bond homolysis (step ii) (Fig. 2c)[35,36]. This high-energy intermediate can undergo either a ring-contraction by 2π-electrocycliczation to azirine **C** (step iii)[26,32,37] or, if R = H, a 1,2-H shift to ketenimine **F** (step vi)[38]. Previous work has suggested that azirine **C** can undergo further photoexcitation to deliver upon isomerization and C−C fragmentation (step iv), the nitrile ylide **D**[38–40]. At this point, ionic cyclization (step v) can complete the isoxazole-to-oxazole isomerization. Regarding the formation of α-ketonitriles **G**, this is most likely arising from a 1,3-H shift of **F** (step vii). The strong difference in reaction performance depending on the position of the Ph-group across the isoxazole core indicates a more complex reaction manifold than previously appreciated.

To understand the differences in photochemical reactivity observed among these three isoxazoles, we carried out computational studies on compounds **2a**, **3a** and **4a** (Fig. 3). Upon photoexcitation to the singlets ($S_1$), all three species undergo N−O bond cleavage via rapid population of a πσ* state[35], independent of the position of the Ph substituent on the isoxazole core. Following internal conversion to the ground state ($S_0$), a singlet biradical is generated (**L2–4**). Previous analysis on this reactivity invoked the formation of a vinylnitrene **B** (see above). While **B** and **L** can be considered as resonance structures,

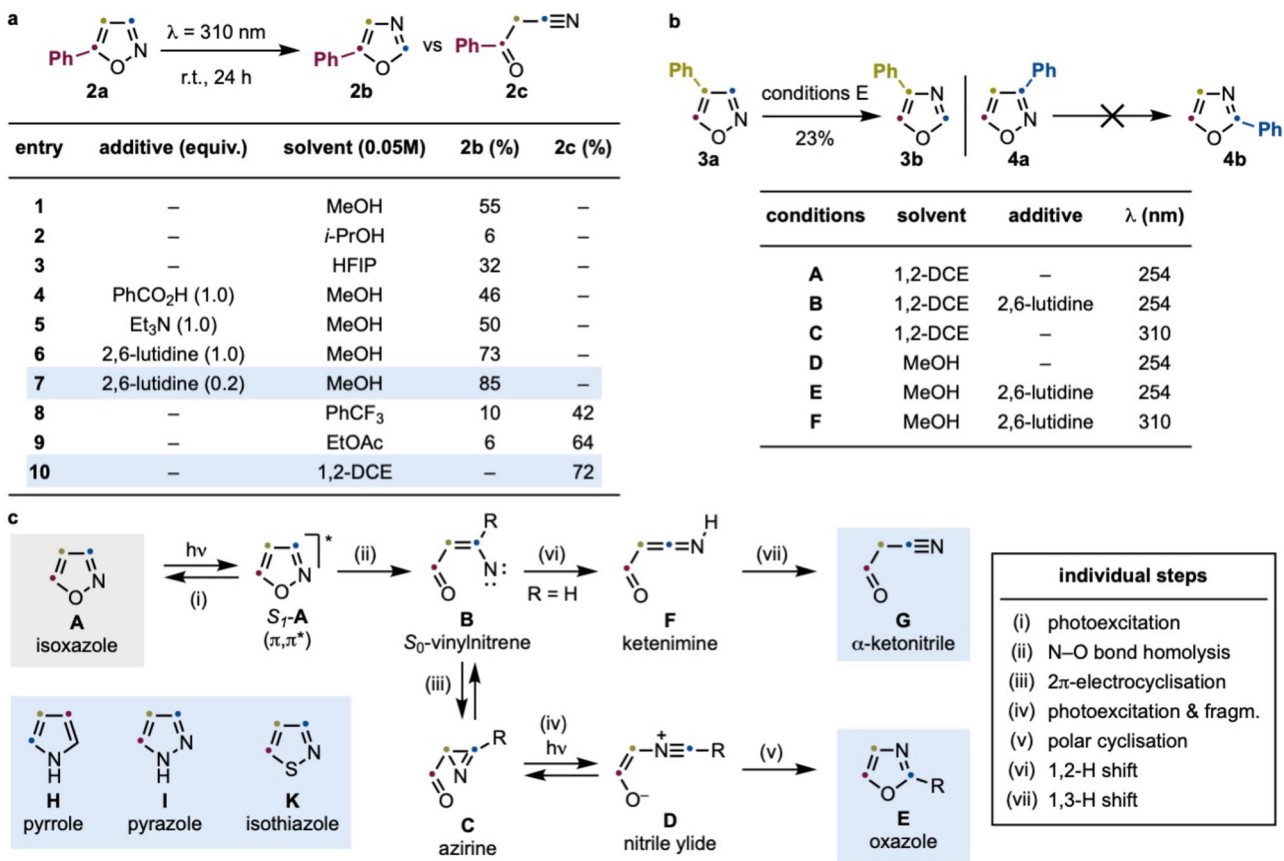

**Fig. 2 | Isoxazole-to-oxazole conversion.** **a** Development of isoxazole-to-oxazole and isoxazole-to-α-ketonitrile photochemical conversions using **1a**. **b** Extension of photochemical reactivity to **2a** and **3a**. **c** Mechanistic analysis of isoxazole photochemistry. $^1$H NMR yields using 1,3-dinitrobenzene as an internal standard are reported.

our calculations suggest **L** as the best way of representing this inter-mediate on the basis of their spin density. From this intermediate, regeneration of the isoxazole or formation of the azirine can occur. For **L4** (blue line), the pathway leading to isoxazole **4a** regeneration is favored, with a significantly lower activation barrier (6.1 kcal mol$^{-1}$, via TS1) compared to that for azirine **C3** formation (18.4 kcal mol$^{-1}$, via TS2). In contrast, for both **L3** (yellow line) and **L2** (red line) derivatives, azirine **C3** and **C2** formation is preferred, as the activation barriers for C–N bond development (**C3**: 2.0 and **C2**: 3.5 kcal mol$^{-1}$, respectively) are lower than those for isoxazole reformation (**3a**: 7.2 and **2a**: 6.0 kcal mol$^{-1}$). The conversion of the azirine intermediates **C2** and **C3** to the corresponding oxazoles **2b** and **3b** via a thermal process is associated with prohibitively high activation barriers (TS3), supporting the hypothesis that this transformation is also photochemically driven. Indeed, our calculations demonstrate that photoexcitation of **C2** and **C3** to the singlet states ($S_1$) enables a facile conversion into the nitrile ylides **N2** and **N3** that can evolve into **2b** and **3b** via low-barrier pro-cesses. To account for the experimental difference in reactivity effi-ciency between the two starting isoxazoles **2a** and **3a**, we speculated that differences in the absorption profile of their corresponding azir-ines might impact the overall photochemical behavior. Notably, **C3** exhibits minimal absorption in the $\lambda = 200–300$ nm range, while **C2** shows strong absorption and should readily undergo photoexcitation. Overall, these computational findings might be used to rationalize the experimental outcomes: **4a** (complete recovery of starting material) undergoes N–O cleavage upon photoexcitation, but the high barrier to azirine formation favors reversion to the starting isoxazole. **3a** (~20% product, ~20% starting material, ~60% decomposition) and **2a** (85% yield) undergo N–O cleavage, with azirine formation being slightly favored. However, due to poor UV absorption, the azirine coming from

**3a**, **C3**, cannot efficiently re-enter the photochemical manifold, leading to competing thermal degradation pathways. In contrast, the azirine intermediate coming from **2a**, **C2**, strongly absorbs in the UV range, enabling a second photoexcitation that drives conversion to the oxa-zole product. The solvent-controlled conversion of **2a** into either **2b** (MeOH) or **2c** (DCE) is more difficult to rationalize. We currently pro-pose that the divergence arises from differential stabilization of reac-tive nitrile ylide intermediates, controlled by solvent polarity and proticity. A plausible explanation for this pronounced solvent effect involves the differential stabilization of the nitrile ylide intermediate. In methanol (MeOH), a polar protic solvent, the nitrile ylide is likely significantly stabilized, shifting the equilibrium toward its formation. This, in turn, facilitates an intramolecular cyclization, ultimately affording the oxazole product. In contrast, 1,2-dichloroethane (DCE), being an apolar aprotic solvent, likely disfavors the formation of the nitrile ylide and instead promotes two consecutive [1,n]-hydrogen shifts, resulting in the formation of the α-ketonitrile product. Overall, these results suggest that the observed experimental selectivity arises not only from the initial photoactivation but also from downstream differences in the photophysical behavior of high-energy intermediates.

Given the strong dependence of reactivity on changes in sub-stitution, we recognized the need for a comprehensive and systematic evaluation of isoxazole substitution patterns (Fig. 4). We sought to explore relevant mono- and disubstituted isoxazoles across a diverse range of aryl (Ph), alkyl (Me), and electron-withdrawing (CO$_2$Me) groups. We therefore selected a set of 24 isoxazole derivatives cate-gorized into five structural classes: (1) Ph-monosubstituted (**2a**–**4a**); (2) ester-monosubstituted (**5a**–**7a**); (3) Ph,Me-disubstituted (**8a**–**13a**); (4) ester,Me-disubstituted (**14a**–**19a**); and (5) Ph,CO$_2$Me-disubstituted

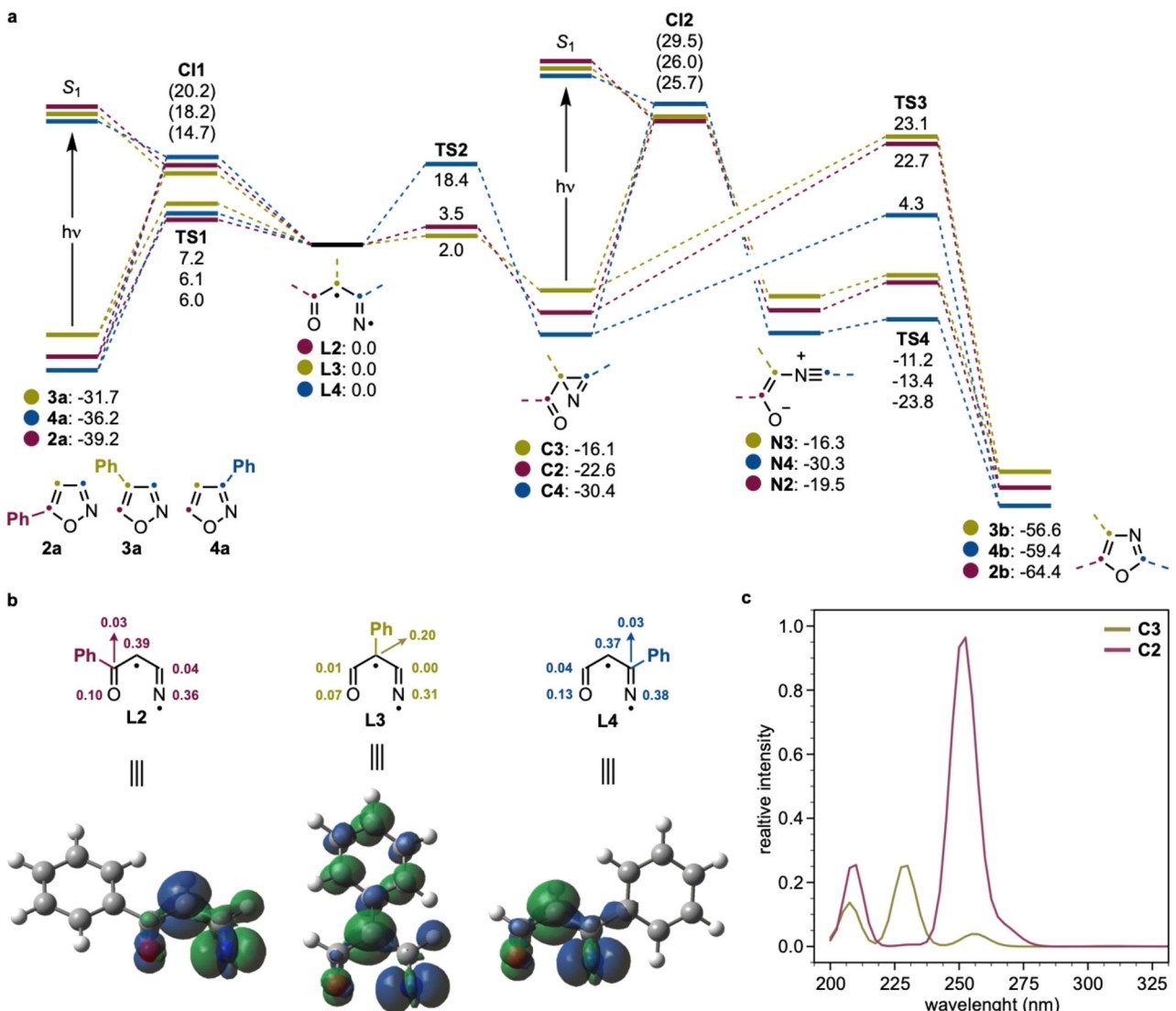

**Fig. 3 | Computational studies. a** Formation of oxazoles **2b**–**3b** from isoxazoles **2a**–**4a** (Gibbs free energies in kcal mol⁻¹ are given related to the corresponding singlet biradical intermediate **Ln**). **b** Calculated spin densities for intermediates **L2**–**L4**. **c** Calculated absorption profiles for intermediates **C2** and **C3**. Computational method: (TD-)CAM-B3LYP/cc-pVTZ/SMD(MeOH)//CAM-B3LYP/cc-pVDZ/SMD(MeOH).

(**20a**–**25a**). Each derivative was subjected to six photochemical conditions (A–E) varying solvent (MeOH vs 1,2-DCE), additive (with or without 2,6-lutidine), and light source ($\lambda = 254$ vs 310 nm). In Fig. 4, only the result leading to the highest yield/lowest decomposition was reported for each substrate; the remaining results can be found in the Supplementary Information Section 4.

Our systematic analysis revealed a highly sensitive reactivity profile dictated by the nature and position of substituents on the isoxazole ring. In many cases, small structural changes dramatically altered photochemical outcomes, ranging from clean isomerization to complete decomposition or photostability, highlighting our inherent low understanding of isoxazole photochemistry. We have performed detailed ¹H NMR spectroscopy analyses of the crude mixtures for representative low-yielding examples. These spectra reveal that aside from the desired oxazoles, the major components typically consist of decomposition products manifesting as broad, unresolved signals in the aromatic and aliphatic regions, consistent with polymeric or oligomeric byproducts. Unfortunately, these byproducts could not be cleanly isolated or fully characterized due to their complexity and instability. In some cases, minor amounts of starting isoxazole were also detected.

For instance, within the Ph-substituted series, only the 5-Ph isomer (**2a**) underwent efficient and selective rearrangement. The 3-Ph (**4a**) was photostable, while 4-Ph (**3a**) gave a low yield of the corresponding oxazole. For ester-substituted isoxazoles, the 3- and 4-substituted variants (**5a** and **6a**) were unstable and decomposed across all tested conditions, precluding rearrangement. In contrast, the 5-substituted isomer (**7a**) cleanly afforded oxazole **7b** in 38% yield, underscoring the privileged reactivity of the 5-position. These findings support the hypothesis that productive rearrangement correlates with the favorability of azirine formation and subsequent re-excitability, as seen in our computational studies.

In the Ph,Me-disubstituted series, most isomers were accessible and evaluated, with the exception of **11a**, whose synthesis remains elusive, highlighting the synthetic limitations even for apparently simple heterocycles. Here, introducing a Me group at C4 (**8a**) enabled reactivity from an otherwise inert 3-Ph core (see the results for **4a**), forming **8b** in moderate yield. However, the 5-Me (**9a**) and 3-Me (**10a**) analogs decomposed entirely. Notably, **12a** and **13a** underwent efficient rearrangement to **12b** and **13b**, respectively, with wavelength sensitivity influenced by the methyl group's position (310 nm for **12a** vs

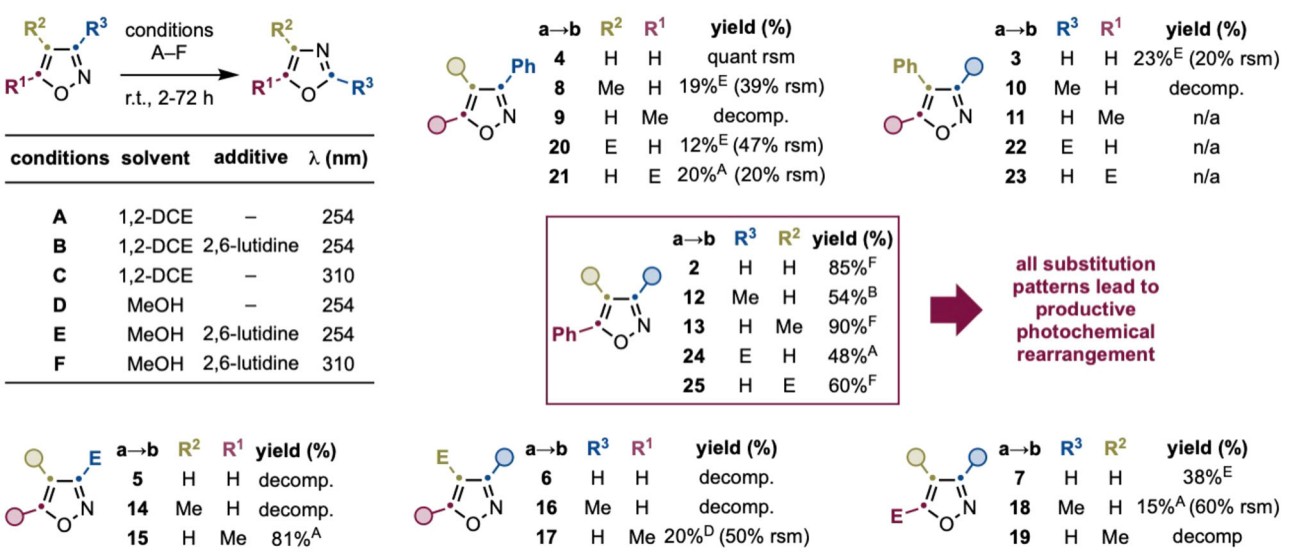

**Fig. 4 | Scope analysis of differentially substituted isoxazoles.** E = $CO_2Me/CO_2Et$. Isolated yields are reported (see Supplementary Information Section 7 for details on recovered starting material for each scope entry).

254 nm for **13a**). The ester,Me-disubstituted isoxazoles, posed greater challenges. Several substrates (**14a**, **16a**, and **19a**) decomposed under irradiation, while **15a**, **17a**, and **18a** yielded the corresponding oxazoles with variable efficiency. In the case of **18a**, careful reaction timing (2 h irradiation time) was key to maximizing yield.

Finally, in the Ph,$CO_2Me$-disubstituted series, both 3-Ph derivatives (**20a** and **21a**) successfully underwent rearrangement to give oxazoles **20b** and **21b** in moderate yields. Although **22a** and **23a** were synthetically inaccessible, the 5-Ph analogs **24a** and **25a** rearranged smoothly to **24b** and **25b** in good yields.

Taken together, this survey identified 5-substituted isoxazoles bearing aryl or electron-withdrawing groups as privileged substrates for selective ring-replacement, while reactivity at other positions proved highly variable and substrate-specific. We therefore focused subsequent synthetic efforts on this class. In terms of synthetic potential, it is interesting to note that this photochemical strategy uses inexpensive and easy-to-make isoxazoles to access 2,5-disubstituted oxazoles (**12b**, **15b**, and **24b**) that in turn are generally difficult to make or very expensive (see the Supplementary Information Section 9 for more details).

We next evaluated the generality of the transformation using a broad range of 5-aryl isoxazoles, which our initial screen identified as robust substrates (Fig. 5a). The reaction conditions (typically $\lambda = 310$ nm, MeOH, 2,6-lutidine) were broadly applicable, with minor tuning required in isolated cases. Overall, the transformation proved tolerant to diverse functional groups.

The method tolerated a wide range of aryl substituents, including electron-donating (Me, OMe, thioether, and $NH_2$; see **26a–29a**) and electron-withdrawing (halides, $CF_3$, CN, and B(pin); see **30a–34a**) groups. Remarkably, sensitive functionalities such as vinyl and $C(sp^2)$–Br groups (see **31b** and **35b**) remained intact, highlighting the method's chemoselectivity.

*Meta-* and *ortho*-substituted arenes (**36a**, **37a**) as well as polyfunctional systems (**38a–40a**) also performed well, demonstrating broad positional tolerance. While the reaction for **41a** was initially low-yielding under standard conditions, extending the reaction time to 72 h at $\lambda = 254$ nm improved the yield to 59%. Importantly, heteroaromatic groups were compatible, including benzoxazole (**42a**), benzothiazole (**43a**), thiophene (**44a**), and furan (**45a**) derivatives, which underwent clean isomerization with no damage to the auxiliary heterocycles.

We also examined polysubstituted isoxazoles (**46a–52a**), which required individual optimization. Trisubstituted systems featuring

combinations of aryl and Me groups rearranged selectively to their oxazoles, provided that substitution patterns aligned with those favoring productive excited-state behavior. The high yield observed for **49b** (from 3-Ph,5-Me isoxazole) contrasts with the decomposition seen for simpler analogs (**9a**, **20a**), again underlining the complex interplay between sterics, electronics, and photophysical properties.

A particularly impactful application was the two-step synthesis of oxazole **50b**, a derivative of a friulimicin B lipopeptide, previously accessed via an eight-step route[41]. Our method reduced this to a simple esterification followed by photochemical rearrangement, showcasing the synthetic streamlining potential of this approach.

Finally, we demonstrated late-stage applicability by converting two pharmaceutically relevant isoxazoles—parecoxib (**51a**) and a Hepatitis C antiviral (**52a**)—into their oxazole analogs (**51b**, **52b**). These transformations illustrate the direct replacement of heterocyclic cores without de novo synthesis, which is otherwise required for SAR studies.

Interestingly, certain substrates exhibited divergent or tandem reactivity, opening new avenues for discovery. Isoxazoles bearing two aryl groups at C4 and C5 (**53a–55a**) underwent both oxazole formation and 6π-electrocyclization, yielding fused polycyclic products with potential applications in organic electronics (Fig. 5b). The selectivity and modularity of this cascade merit further exploration in materials-oriented contexts[42,43].

Even more striking was the behavior of **12a**, which could be selectively diverted to three different products depending on conditions (Fig. 5c). While conditions B (DCE, 2,6-lutidine, $\lambda = 254$ nm) gave oxazole **12b**, reactions in MeOH yielded **12c** via trapping of a ketenimine intermediate[44], and lower-energy irradiation in DCE afforded the azirine **12 d**. This conditional divergence from a single scaffold underscores the unique power of photochemical control and further validates the need for detailed reactivity mapping.

As discussed above, while investigating the isoxazole-to-oxazole rearrangement, we discovered that solvent tuning can redirect the reaction pathway toward C–C bond cleavage, yielding α-ketonitrile intermediates. Notably, α-ketonitriles are valuable intermediates in condensation chemistry and can also be accessed by conventional routes. However, their direct formation from isoxazoles offers a streamlined and synthetically powerful "ring-replacement" opportunity to directly convert them into pyrroles (**H**), pyrazoles (**I**), and isothiazoles (**K**).

It is important to note that, at present, the photochemical isoxazole-to-α-ketonitrile conversion is only accessible using 5-Ar-

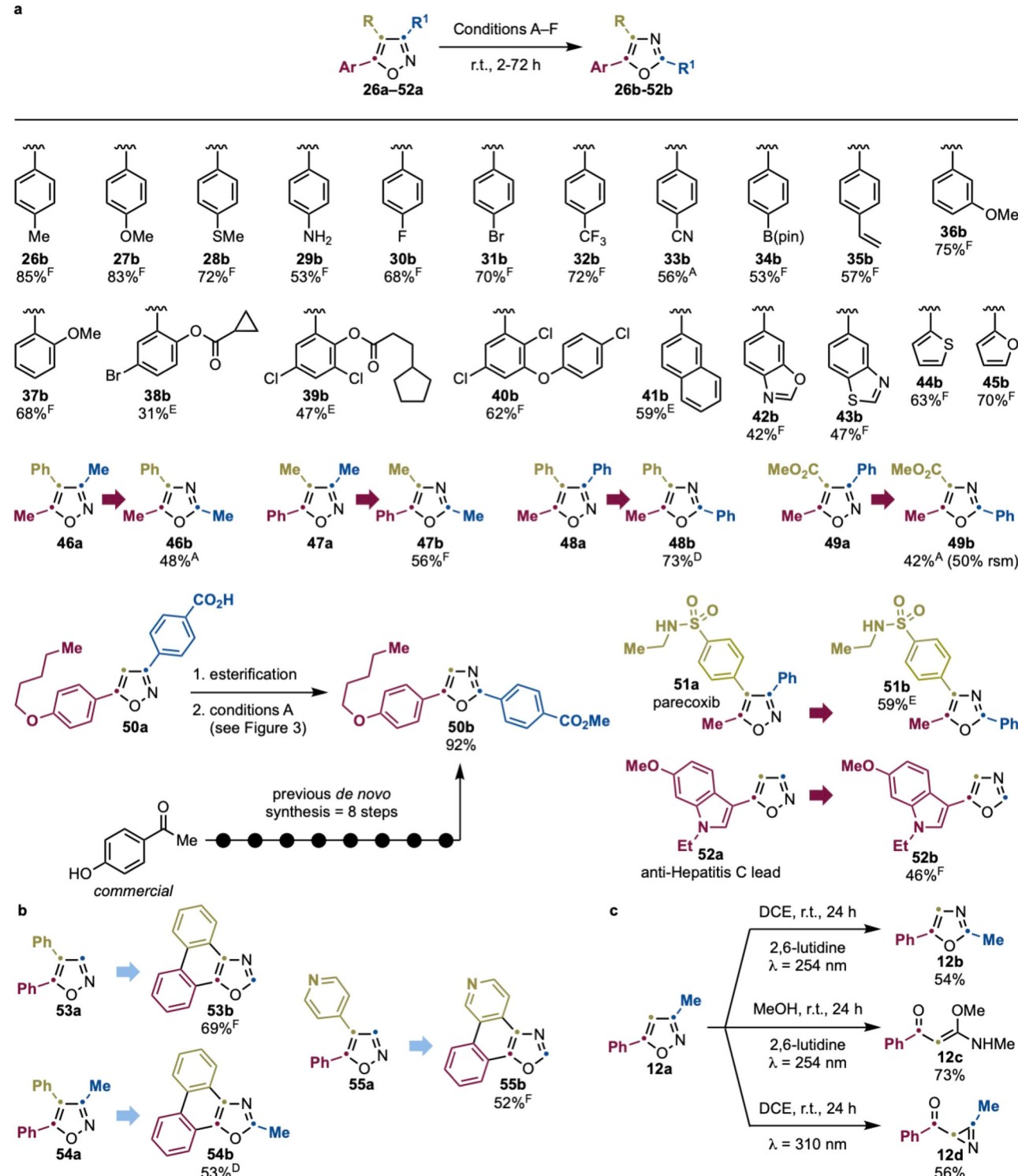

**Fig. 5 | Photochemical isomerization of isoxazoles to oxazoles. a** Scope of the process. **b** Concomitant isoxazole-to-oxazole conversion and 6π electrocyclization. **c** Different photochemical behavior of **12a**. Isolated yields are reported (see Supplementary Information Section 7 for details on recovered starting material for each scope entry). Reaction conditions: [A] 1,2-DCE, r.t., 24–48 h, λ = 254 nm; [B] 2,6-lutidine, 1,2-DCE, r.t., 24–48 h, λ = 254 nm; [C] 1,2-DCE, r.t., 24–48 h, λ = 310 nm; [D] MeOH, r.t., 24–48 h, λ = 254 nm; [E] 2,6-lutidine, MeOH, r.t., 24–48 h, λ = 254 nm; [F] 2,6-lutidine, MeOH, r.t., 24–48 h, λ = 310 nm.

substituted derivatives. The inclusion of other substituents on the isoxazole core leads mostly to the formation of oxazoles, photostability, or photodecomposition (see Figs. 3 and 4).

Recognizing the synthetic potential of α-ketonitriles, we designed a modular platform for isoxazole-based scaffold remodeling, converting these intermediates into five heterocyclic frameworks via one-

pot transformations (Fig. 6a, b). Specifically, in order to demonstrate the applicability of this approach in generating a large library of heteroaromatic derivatives, we identified conditions for converting 5-aryl-isoxazoles into five distinct heterocyclic systems: amino-pyrazoles, pyrazoles, pyrroles, amino-isoxazoles, and isothiazoles. This methodology was applied to a selection of seven commercially available

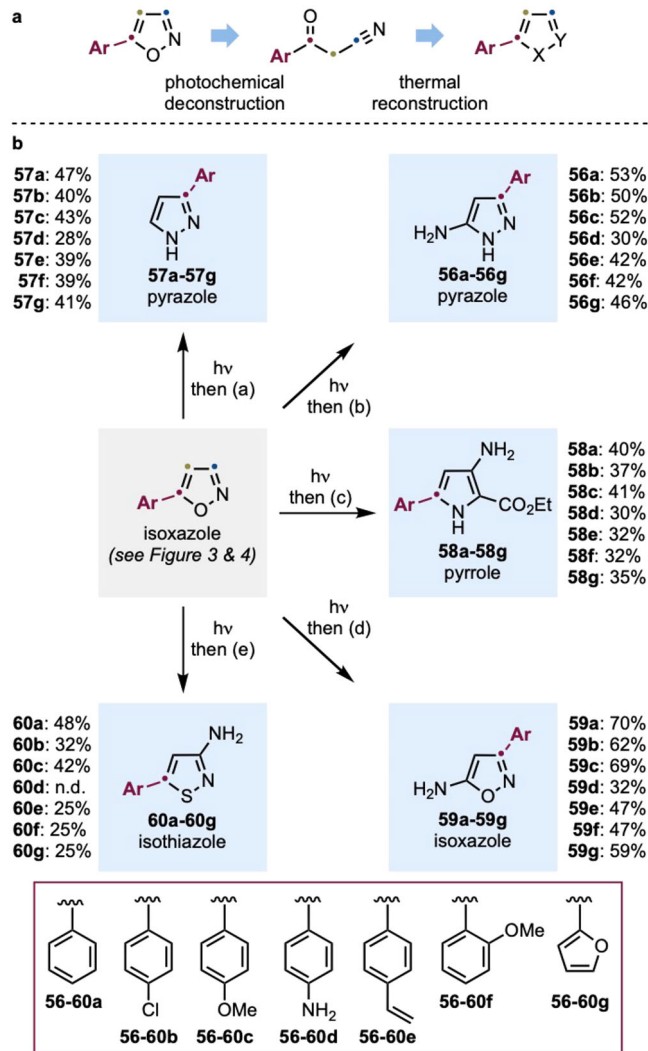

**Fig. 6 | Direct and divergent isoxazole "ring-replacement". a** Photochemical deconstruction–reconstruction strategy for divergent "ring-replacement of isoxazoles". **b** Development of a library on isoxazole "ring-replacement. Reaction conditions: hv = 1,2-DCE, r.t., 24–48 h, $\lambda$ = 310 nm. (a) PhSO$_2$NHNH$_2$, NBS, EtOH, 80 °C. (b) DIBAL-H, Et$_3$N, THF, r.t. then NH$_2$NH$_2$•H$_2$O, *i*-PrOH, 70 °C. (c) TsCl, Et$_3$N, CH$_2$Cl$_2$, r.t., then diethyl aminomalonate, EtONa, EtOH:THF, r.t. (d) NH$_2$OH•HCl, MeOH, rt. (e) Mukaiyama reagent, Et$_3$N, CH$_2$Cl$_2$, r.t. then Na$_2$S, ClNH$_2$, EtOH:H$_2$O, 70 °C. Isolated yields are reported (see Supplementary Information Section 7).

starting materials (**2a, 27a, 29a, 35a, 37a, 45a**, and the 4-Cl-Ph not represented in Fig. 6), delivering a library of 34 new derivatives without resorting to de novo individual syntheses.

Each substrate was first subjected to the optimized photochemical conditions (DCE, $\lambda$ = 310 nm), generating the corresponding α-ketonitriles, which were immediately used in subsequent condensation reactions.

For instance, treatment with benzenesulfonyl hydrazide in EtOH followed by heating under reflux enabled isoxazole-to-3- amino-pyrazole "ring-replacement" (**56a–56g**). The 5-amino-pyrazole motif is highly versatile, with widespread applications in drug discovery, particularly in anticancer, antibacterial, antimalarial, and anti-inflammatory agents[7,45].

Alternatively, DIBAL-H reduction of the α-ketonitriles, followed by condensation with H$_2$N–NH$_2$•H$_2$O, converted 5-aryl-isoxazoles into 3-aryl-pyrazoles **57a–57g**[46]. This transformation represents a formal oxygen-to-nitrogen transmutation, and while related strategies are known, they typically require nickel

catalysts and high-pressure hydrogenation, making our method a milder alternative[47–49].

Further heterocyclic diversity was achieved by treating the crude α-ketonitriles with tosyl chloride and then diethyl aminomalonate to form 3-amino-pyrroles (**58a–58g**)[50].

Additionally, refluxing the crude α-ketonitriles with HO–NH$_2$•H$_2$O in EtOH led to the conversion of 5-aryl-isoxazoles into 3-aryl-5-amino-isoxazoles (**59a–59g**)[51]. This transformation effectively moves the aromatic substituent from C5 to C3 while introducing an amino group at C5.

Lastly, the isoxazole-to-isothiazole conversion was realized as a formal O-to-S atom transmutation. This was achieved by converting the α-ketonitriles into an alkyne using the Mukaiyama reagent, followed by reaction with sodium sulfide and chloramine-T, furnishing the desired isothiazoles (**60a–60g**) in moderate yields[52].

In summary, we reported a photochemical strategy that transforms isoxazoles into structurally distinct heterocycles, including oxazoles, pyrazoles, pyrroles, and isothiazoles. This platform exploits both ring-contraction and deconstruction–reconstruction pathways, enabled by selective control of excited-state reactivity. The approach provides a rare example of scaffold editing for aromatic heterocycles and offers a practical route to heterocycle diversification without de novo synthesis. Its operational simplicity, broad functional group tolerance, and successful application to drug-like scaffolds suggest strong potential for adoption in medicinal chemistry.

## Methods

### General procedure for the permutation of isoxazole

A dry tube equipped with a stirring bar was charged with the corresponding isoxazole (1.0 equiv.). The tube was capped with a Supelco aluminum crimp seal with septum (PTFE/butyl), evacuated and refilled with N$_2$ (×3). The corresponding anhydrous and degassed solvent and the corresponding additive (0.2 equiv.) were added. The tube was placed into a Helios photoreactor. The photoreactor and a fan were switched on, and the mixture was stirred under irradiation for the specified time. The solvent was evaporated, and the residue was purified by column chromatography on silica gel to give the desired product. All modifications regarding the solvent and/or additive used are detailed in the Supplementary Information in Section 2, page S6.

### General procedure for the preparation of α-ketonitrile

A dry tube equipped with a stirring bar was charged with the corresponding isoxazole (0.2 mmol, 1.0 equiv.). The tube was capped with a Supelco aluminum crimp seal with septum (PTFE/butyl), evacuated and refilled with N$_2$ (×3). Degassed DCE (4 ml) was added. The tube was placed into a Helios photoreactor equipped with 310 nm lamps, and the fan was switched on. The mixture was stirred under irradiation for 24 h. Upon completion, the solvent was evaporated under reduced pressure, and the crude product was used without any further purification.

## Data availability

All data is available in the main text or the Supplementary Information. All data are available from the corresponding author upon request. The source data are provided with the article. Source data are provided with this paper.

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

## Acknowledgements

D.L. thanks the ERC for a grant (101086901). Y.X. thanks Shenzhen University for funding. B.R. acknowledges Janssen for a PhD CASE Award.

## Author contributions

D.L., A.R., and B.R. designed the project. Y.X., L.P., and B.R. run all the synthetic experiments; E.M.A. run all the computational studies. All authors discussed the results and wrote the manuscript.

## Funding

## Competing interests

The authors declare no competing interests.
