## [Transparent Peer Review file · Nature Communications]

Divergent Photochemical Ring-Replacement of Isoxazoles

Corresponding Author: Professor Daniele Leonori

Version 0:

Reviewer comments:

Reviewer #1

(Remarks to the Author)

Leonori and co-workers report in this manuscript a comprehensive study pertaining to the photochemical isomerisation of isoxazoles into oxazoles and related products which typically proceeds via azirines, keto nitriles or ketenimine intermediates. These reactions are well described in the literature for both isoxazoles and analogous heterocycles possessing a weak X-Y bond (for instance as reviewed by Hoffmann: doi.org/10.1002/ejoc.201901190). A recent paper by the authors in Nature highlights the underlying reactivity for related azoles (Nature volume 637, pages860–867 (2025)). While recent years have witnessed a renaissance of this classical transformation in view of 'molecular editing' approaches, the underlying chemistry is well documented for almost 50 years. The authors provide thus a comprehensive summary of the current understanding of these isoxazole-based transformations by varying the substitution patterns as well as electronic properties of the resulting substrates which shows that some building blocks are very suited to undergoing this photo-process whereas low yields as well as significant amounts of decomposed material are observed in other cases. Provision of DFT data aids in understanding this behaviour which follows on from previous computations though these may not have been as comprehensive. As a consequence, the authors report instances where oxazoles are favoured as well as other circumstances/conditions where keto nitriles or transient ketenimines are favoured. This diverse reactivity is very appealing, but recent studies by Bracken and Baumann have already reported these outcomes and provided mechanistic reasoning governing these outcomes (ref 31/43). An interesting addition provided by the authors concerns the use of aryl isoxazoles that can be directed to a variety of different azoles exploiting the ability to bias the formation of a major photo-product that can be converted into pyrazoles, pyrroles as well as isothiazoles/isoxazoles using appropriate reagents and conditions as known in the literature.

It would have been useful if the authors had discussed in more detail what the crude product composition is for cases where only modest amounts of oxazoles are formed, i.e. what other species (and in which amounts) were observed. Does the analysis of these crude mixtures provide new insights in this chemistry? Also, it appears that these reactions are often rather slow (sometimes 72 hours are needed) - why is this? Is it related to the low optical power of the light sources used or are the reaction set-ups photon-limited? Please discuss/demonstrate if this chemistry can be scaled to make gram quantities of valuable products as this would be required to validate the chosen argument that this method is relevant to medicinal and industrial chemists. As a minor point, I didn't understand why several of the UV-Vis spectra (SI45 etc.) show areas of negative absorbance - is that a measurement issue?

Altogether, this is a well-executed study that highlights the diverse reactivity of isoxazoles under photochemical conditions. The manuscript and SI are very informative. My main concern would be in relation to the novelty of this chemistry where I encourage the authors to make it clear to the reader where new understanding on this intriguing reactivity has been found.

Reviewer #2

(Remarks to the Author)

Skeletal editing has emerged as a highly promising area in synthetic organic chemistry, offering substantial potential across various fields, particularly in medicinal chemistry. This strategy provides a shortcut to late-stage core modifications, avoiding de novo synthesis and thereby significantly reducing resource and time consumption. In this context, Leonori and coworkers present an excellent example by remodeling isoxazoles into diverseazole derivatives, with potential applications in lead optimization and drug discovery. Although the photochemical transformation of isoxazoles into corresponding oxazoles has been previously reported by Ulman, Singh, and others, the present study explores the influence of substituents, irradiation wavelengths, and solvent effects, enhancing reaction control and broadening the applicability of the methodology. Overall, I

recommend the publication of this manuscript in Nature Communications after addressing the following points:

- 1) The computational section is brief and lacks sufficient mechanistic detail. The authors are encouraged to provide full information on the computed transition states, particularly TS3 (Fig. 2d). Is the conversion from azirine to oxazole a concerted single-step process, or does it involve intermediates? The methodology for obtaining the absorption profiles of intermediates C2 and C3 should also be clarified (experimental or TD-DFT calculations?). The explanation for the differential reactivity of 3a and 2a, based primarily on absorption profiles, appears insufficient, especially considering the lower yield observed for 2a at 254 nm. A more comprehensive computational analysis would significantly strengthen the mechanistic insight.
- 2) The observed solvent effects are intriguing but insufficiently rationalized. The authors are encouraged to propose plausible mechanistic explanations or hypotheses, supported by experimental or computational evidence, to account for solvent-dependent reactivity. In particular, the behavior of substrates 2a and 12a under different solvents and substitution patterns deserves further discussion.
- 3) The mechanistic rationale behind the effect of 2,6-lutidine on yield improvement remains unclear. Although the authors cite related work by Taniguchi et al., a more detailed discussion is needed to explain the observed increase in yield from 55% to 85%. It would also be helpful to explore the effect of lower equivalents (e.g., 0.1 eq), given that 0.2 eq outperformed higher loadings.
- 4) While the substrate scope is broad, discussing the method's limitations would enhance its practical relevance. Specifically, it would be valuable to explore common functional groups in drug-like molecules that may not tolerate the reaction conditions (e.g., amides, lactones) and to include examples of suboptimal or unsuccessful transformations.
- 5) If feasible, applying the divergent isoxazole ring placement (Fig. 5) to more complex molecules such as substrates 50, 51, or 52 would provide better evidence of the method's generality and synthetic utility.
- 6) The introduction would benefit from better contextualizing this work within the field of skeletal editing, which has seen significant recent advances. The authors should consider citing key recent reviews, such as: Sarpong and Levin, *Nat. Synth.*, 2022, 1, 352–364; Ball, *Helv. Chim. Acta*, 2023, 106, e202200182.
- 7) Although a minor issue, the current compound numbering system may cause confusion. It may be clearer to assign unique numbers to distinct skeletons and use letters (a, b, c, etc.) for derivatives.
- 8) Please consider the commented PDF attached.

Reviewer #3

(Remarks to the Author)

Leonori et al. present a photochemical platform for the selective conversion of isoxazoles into five-membered heteroaromatics by leveraging excited-state reactivity. The authors systematically investigated the effects of substitution sites during the isoxazole-to-oxazole transformation using DFT calculations. Concurrently, they conducted a scope analysis of differentially substituted isoxazoles. For the divergent "ring replacement" of isoxazoles, the key lies in the transient formation of common intermediate α -ketonitriles. By subjecting these intermediates to distinct reaction conditions, the authors observed divergent outcomes, enabling the construction of five heterocyclic frameworks through a one-flask operation.

My main concerns with this manuscript arise from several critical scientific considerations, which cumulatively cast doubt on its readiness for publication in Nature Communications. The concerns can be systematically categorized as follows:

1. The manuscript's core focus on the isoxazole-to-oxazole transformation, while methodologically sound, is circumscribed by critical precedents. As detailed in References 24 and 25 (Ullman, Singh et al.), this rearrangement class has established synthetic precedence, with Reference 31 recently reporting an isoxazole-derived oxazole synthesis under Hg-lamp irradiation. These prior disclosures inherently diminish the work's claimed novelty, despite the authors' contributions to substrate scope expansion and mechanistic elaboration. The study thus manifests as a synthetic and mechanistic refinement rather than an innovation meeting the elevated standards expected of Nature Communications.
2. On page 4, line 13, the authors note that "the 3-phenyl analogue (4a) proved photochemically inert under all tested conditions, even upon prolonged irradiation," with DFT calculations attributing this to a higher energy barrier for aziridine formation. However, a discrepancy arises: substrates 8a, 20a, 21a, 48a, and 49a—all 3-phenyl-substituted isoxazoles—react smoothly under the protocol. This inconsistency necessitates mechanistic clarification. The authors are expected to perform additional DFT calculations to rationalize the reactivity differences between di-substituted and tri-substituted isoxazoles.
3. Additionally, the observed substrate specificity warrants concern. As illustrated in Figure 3, minor substituent variations on the isoxazole scaffold lead to divergent reaction outcomes. Even for privileged aryl substrates bearing 5-substituted isoxazole moieties (e.g., 33a, 38a, 41a), reaction conditions required case-by-case optimization to enable the transformation. Such results undermine the synthetic utility of the reported method.
4. The SI is well-prepared, but ¹H NMR spectra of 43b and 58d show impurities. Please re-purify and resubmit clean spectra.

Reviewer #4

(Remarks to the Author)

Reviewer #5

(Remarks to the Author)

The manuscript presents a novel photochemical platform for the transformation of isoxazoles into structurally diverse heterocycles, including oxazoles, pyrazoles, pyrroles, and isothiazoles. This work provides significant value for the derivatization of five-membered nitrogen-containing heterocycles in pharmaceutical development. Although a robust methodological design, comprehensive data collection, and detailed DFT calculations are done to investigate the complex interplay between sterics, electronics, and photophysical properties, certain aspects of the mechanistic analysis and discussion require further clarification.

(1) The proposed mechanism leading to the isoxazole-to-oxazole (A-to-E) isomerization includes the photoexcitation of isoxazole A, N–O bond homolysis, 2π -electrocyclicization, and second photoexcitation to produce the corresponding oxazoles. To better understanding the selectivity, the energies of the excited singlet state (S_1 -A) and the corresponding conical intersection (CI, S_1/S_0) points should be determined in the energy profiles.

(2) According to the mechanism in Figure 2c, the conversion from C to E involves second photoexcitation and C-N bond cleavage to obtain nitrile ylide (D), and then through an electrocyclicization transition state to obtain the oxazole product. However, the second photoexcitation process and the detailed structure and energy information of the nitrile ylide (D) are missing in Figure 2d. As reported in Su's work (J. Phys. Chem. A 2015, 119, 9666–9669), it is separated processes for C-N bond cleavage (C to D) and C-O bond formation (D to E). Is TS3 a concerted process for these two steps or just C-O bond formation from nitrile ylide?

(3) The potential for intersystem crossing (ISC) from singlet to triplet along the reaction pathways should be considered, including the triplet states of isoxazole (T_1 -A), and the triplet states of other key intermediates, just in case the involvement of triplet-state mechanisms.

(4) The NMR spectral data require the following corrections: The ^1H NMR integrals need calibration, as exemplified by the unlabeled integrations for peaks 8a and 18a. Additionally, the notation ' ^{13}C NMR' for compound 52a (151 MHz, CDCl_3) should be revised to ' ^{13}C NMR.' Furthermore, the ^1H NMR integrals for peaks 39a/b must be properly adjusted for accuracy.

Major Revision The authors should address the above concerns, particularly the mechanistic clarifications and the explanation for the energy barrier/yield discrepancy, before the manuscript can be reconsidered for publication. The revisions are essential for ensuring the robustness and clarity of the proposed mechanism.

Version 1:

Reviewer comments:

Reviewer #2

(Remarks to the Author)

The manuscript has improved, and I appreciate the authors' efforts. However, I still have two major concerns: (i) the synthetic value/generalizability remains uncertain, as small substituent changes seem to cause large outcome shifts and performance on complex, drug-like substrates appears limited; and (ii) the mechanistic basis of the differential reactivity of 3a and 2a leans mainly on TD-DFT absorption profiles of intermediates (not a solid evidence and doesn't explain the whole reaction profile), and does not yet offer a solid explanation for solvent/additive effects.

Reviewer #3

(Remarks to the Author)

The authors have addressed most of my concerns, yet some aspects still require further refinement.

Below are my comments on the resubmitted manuscript.

The authors stated that "Small changes in substitution patterns (e.g., replacing a Ph with a Me or CO_2Me) can dramatically alter excited-state behaviour, reaction pathways, and product distributions." While it is reasonable to anticipate the need for reaction condition adjustments when a phenyl group is substituted with a methyl or ester group—given the distinct electronic and steric properties of these moieties—the fact that 33a, 38a, and 41a all fall within the 5-aryl isoxazole scaffold undermines the persuasiveness of this explanation.

In the ^1H NMR spectrum of compound 58d, both solvent peaks and impurity peaks are non-negligible, as they can significantly interfere with the accurate determination of the compound's purity—and thus indirectly affect the reliable calculation of its yield. If further purification proves unfeasible, this substrate should be excluded from subsequent studies.

Additionally, after carefully reviewing the authors' responses to Reviewer #1, this reviewer confirms that the latter's technical requests have been properly addressed.

Reviewer #4

(Remarks to the Author)

Reviewer #5

(Remarks to the Author)

In the new version, the authors did a more detailed mechanistic study, both in calculation and experimentally. As shown in Figure 3, the second photoexcitation process and the detailed structure and energy information of the nitrile ylide are calculated. The energies of the S1/S0 conical intersections are also added to support the proposed mechanism. The triplet-state mechanisms are excluded by EnT catalysis screening experiments using a range of known triplet photosensitizers, but if the authors have calculated the energies of triplet states of the key intermediates. Are they also consistent with the experimental results?

After the author's efforts, most of my concerns have been addressed in the new revisions. The revised manuscript could be considered for publication.

Version 3:

Reviewer comments:

Reviewer #5

(Remarks to the Author)

I am ok with the manuscript and don't have further comments.

Reviewer #1: Leonori and co-workers report in this manuscript a comprehensive study pertaining to the photochemical isomerisation of isoxazoles into oxazoles and related products which typically proceeds via azirines, keto nitriles or ketenimine intermediates. These reactions are well described in the literature for both isoxazoles and analogous heterocycles possessing a weak X-Y bond (for instance as reviewed by Hoffmann: doi.org/10.1002/ejoc.201901190). A recent paper by the authors in Nature highlights the underlying reactivity for related azoles (Nature volume 637, pages860–867 (2025)). While recent years have witnessed a renaissance of this classical transformation in view of 'molecular editing' approaches, the underlying chemistry is well documented for almost 50 years. The authors provide thus a comprehensive summary of the current understanding of these isoxazole-based transformations by varying the substitution patterns as well as electronic properties of the resulting substrates which shows that some building blocks are very suited to undergoing this photo-process whereas low yields as well as significant amounts of decomposed material are observed in other cases. Provision of DFT data aids in understanding this behaviour which follows on from previous computations though these may not have been as comprehensive. As a consequence, the authors report instances where oxazoles are favoured as well as other circumstances/conditions where keto nitriles or transient ketenimines are favoured. This diverse reactivity is very appealing, but recent studies by Bracken and Baumann have already reported these outcomes and provided mechanistic reasoning governing these outcomes (ref 31/43). An interesting addition provided by the authors concerns the use of aryl isoxazoles that can be directed to a variety of different azoles exploiting the ability to bias the formation of a major photo-product that can be converted into pyrazoles, pyrroles as well as isothiazoles/isoxazoles using appropriate reagents and conditions as known in the literature.

We thank the Reviewer for their thorough assessment and for raising important points regarding the novelty of our work. We would like to bring some clarifications what we hope will be taken into consideration.

- **Historical context.** We agree that the photochemistry of isoxazoles has some important precedents. However, these reports are fragmented, often limited to a handful of substrates, and lack mechanistic or predictive guidelines. For example, Pavlik's study (ref. 27) examined only five substrates, with preparative data for just two. The remaining cases were based on GC analysis only, without product isolation or characterization, limiting broader conclusions.
- **Selectivity.** In their work, Pavlik also used substrate **2a** and reported a mixture of oxazole **2b** (42%) along α -ketonitrile **2c** (27%). By contrast, in our study, we can selectively obtain either **2b** or **2c** in >80% yield by changing the reaction conditions (see table below). We believe this represents a significant advance with strong synthetic implications. A similar situation applies to other examples where Pavlik's outcomes are mixtures – please note that their data (e.g. **12a** to **12b**) is not based on synthetic experiments but rather on GC conversion based on recovered starting material.

	2a	2b	2c
State-of-the-art			
		2b	2c
J. Het. Chem. 2005, 273		42	27
			rsm
			4
Our work			
	Conditions C	–	84
	Conditions F	85	–

- **Scope and mechanism.** Our work systematically surveys a wide range of substitution patterns, revealing dramatic and unexpected reactivity differences depending on each substituent and, more remarkably, their position on the isoxazole core. This illustrates that reactivity trends established for a single class of substrates (e.g., Bracken/Baumann with C3-ester, C5-aryl isoxazoles) cannot be generalized across the isoxazole family and do not represent the complexity of the chemistry.

We show that variations in reactivity often stem from changes in the absorption profiles of reactive intermediates – an effect not previously considered. Our combined experimental/DFT analysis provides mechanistic rationales missing from earlier studies.

In summary, while we build our work upon important prior contributions, these earlier reports feature limited scope, often non-preparative conditions, low or unreported yields, and no general strategy for controlling product distribution. I hope the Reviewer and the Editor will consider our study delivering the first systematic framework that enables condition-dependent, selective access to divergent products from some isoxazole substrates. We believe this constitutes a substantive advance with strong implications for synthetic chemistry and photochemistry.

It would have been useful if the authors had discussed in more detail what the crude product composition is for cases where only modest amounts of oxazoles are formed, i.e. what other species (and in which amounts) were observed. Does the analysis of these crude mixtures provide new insights in this chemistry?

We have performed detailed ^1H NMR spectroscopy analyses of the crude mixtures for representative examples featuring low-yields (e.g. **3a**, **8a**, **9a** and **10a** in Figure 3). These spectra reveal that aside from the desired oxazoles, the major components typically consist of decomposition products manifesting as broad, unresolved signals in the aromatic and aliphatic regions, consistent with polymeric or oligomeric byproducts. Unfortunately, these byproducts could not be isolated and characterized so we cannot propose a tentative structure and decomposition pathway. In some cases, minor amounts of starting isoxazole were detected.

Also, it appears that these reactions are often rather slow (sometimes 72 hours are needed) - why is this? Is it related to the low optical power of the light sources used or are the reaction set-ups photon-limited?

We do not think our reaction set up to have low optical power as the light bulbs we use are powerful. We believe the second option mentioned by the Reviewer to be more likely. Unfortunately, we do not have the equipment for measuring quantum yields for these reactions.

Please discuss/demonstrate if this chemistry can be scaled to make gram quantities of valuable products as this would be required to validate the chosen argument that this method is relevant to medicinal and industrial chemists.

We have been able to scale-up the preparation of **2b** on 8.0 mmol scale without any significant change in yield. This result has been added to the revised manuscript and SI.

As a minor point, I didn't understand why several of the UV-Vis spectra (SI45 etc.) show areas of negative absorbance - is that a measurement issue?

We tried several times to rerun the measurements, but we have not been able to remove the areas of negative absorbance, which upon discussion with the technical support of our UV/Vis instrument are believed to be artifacts.

Reviewer #2: Skeletal editing has emerged as a highly promising area in synthetic organic chemistry, offering substantial potential across various fields, particularly in medicinal chemistry. This strategy provides a shortcut to late-stage core modifications, avoiding de novo synthesis and thereby significantly reducing resource and time consumption. In this context, Leonori and coworkers present an excellent example by remodeling isoxazoles into diverseazole derivatives, with potential applications in lead optimization and drug discovery. Although the photochemical transformation of isoxazoles into corresponding oxazoles has been previously reported by Ulman, Singh, and others, the present study explores the influence of substituents, irradiation wavelengths, and solvent effects, enhancing reaction control and broadening the applicability of the methodology.

Overall, I recommend the publication of this manuscript in Nature Communications after addressing the following points:

1) The computational section is brief and lacks sufficient mechanistic detail. The authors are encouraged to provide full information on the computed transition states, particularly TS3 (Fig. 2d).

We have performed additional computational work to fully elucidate the reaction path. The various **TS3s** correspond to the thermal isomerization of azirines **C2–4** to oxazoles **2b–4b**, and do not reflect any potential photochemical pathway. These thermal reactions are concerted yet asynchronous: the C–C cleavage occurs first, followed by C–O bond formation. Since these barriers are energetically costly, we propose that photoexcitation of the azirines might overcome them and lead to the oxazoles via the nitrile ylides **N2–N4**. The energy barriers that separate these ylides from their corresponding oxazoles are low, so these species should evolve fast to the formation of the final products. This pathway helps also rationalise the difference in reactivity of isoxazoles **2a** and **3a**, since their corresponding azirines **C2** and **C3** have drastically different absorption profiles. We revised our discussion in the revised manuscript. Furthermore, we have included the energies of the S_1/S_0 conical intersections for all species and for both photochemical pathways.

Is the conversion from azirine to oxazole a concerted single-step process, or does it involve intermediates? The methodology for obtaining the absorption profiles of intermediates C2 and C3 should also be clarified (experimental or TD-DFT calculations?).

The absorption spectra were calculated using TD-DFT, specifically at the TD-CAM-B3LYP/cc-pVTZ/SMD level of theory. This was already indicated in the original version of the SI and we have also added it to the caption for Figure 2d.

The explanation for the differential reactivity of 3a and 2a, based primarily on absorption profiles, appears insufficient, especially considering the lower yield observed for 2a at 254 nm. A more comprehensive computational analysis would significantly strengthen the mechanistic insight.

We have modelled the full pathway from isoxazole to oxazole considering both thermal and photochemical steps and our proposed explanation for the reactivities of **2a** and **3a** is based on the combination of all these

computational results. It is difficult to consider other aspects to compute beyond the ones we have already discussed. Furthermore, both compounds **2a** and **3a** have the same thermal barriers for the N–O cleavage/regeneration (TS1), for the formation of the azirine from the vinylnitrene (TS2), and for the formation of the oxazole from the azirine (TS3). The only difference we have been able to find is in the absorption spectra of their azirine intermediates; **C2** (the azirine form of **2a**) has a strong absorption while **C3** (the azirine form of **3a**) does not. If the Reviewer has some specific aspects or suggestion in mind, we would be very happy to consider them.

2) The observed solvent effects are intriguing but insufficiently rationalized. The authors are encouraged to propose plausible mechanistic explanations or hypotheses, supported by experimental or computational evidence, to account for solvent-dependent reactivity. In particular, the behavior of substrates **2a** and **12a** under different solvents and substitution patterns deserves further discussion.

We agree with the Reviewer this is one of the most fascinating yet puzzling aspect of permutation chemistry. Indeed, this is not restricted to the isoxazole-to-oxazole conversion but also it played a prominent role in the development of the thiazole-isothiazole permutations. However, dissecting the role(s) of the various solvents on such complex reaction profiles where multiple species are involved in both thermal and photochemical steps is very challenging.

We have attempted the calculation of the reaction pathways for the formation of **2b** and **2c** (stopping at the ketenimine intermediate) using an implicit solvation model based on MeOH and DCE solvents. The results, shown below, indicate that the relative barriers for the formation of the azirine and ketenimine intermediates do not change substantially with the solvent polarity, which does not explain the experimental observations. This might be a limitation in the use of implicit solvation. Capturing these effects would require explicit solvation (microsolvation) models, which are highly demanding and, we believe, beyond the scope of the present study. We therefore restrict ourselves here to presenting the experimental observations and noting this solvent effect as an open question for future dedicated computational investigations.

Regarding **2a**, we could propose the participation of MeOH, trapping the ketenimine **F** and preventing the last hydrogen shift forming the α -ketonitrile. Regarding **12a**, the absence of hydrogen at C3 precludes the formation of the α -ketonitrile and therefore, in the presence of MeOH, the unstable ketenimine intermediate is trapped and forms product **12c** (see *Org. Lett.* **2023**, *25*, 6593–6597 for the formation of this ketenimine intermediate). In the absence of such nucleophile, the ketenimine could revert to the vinylnitrene **B** and then follow the other mechanistic pathway leading to either azirine **C** at 310 nm (given azirines **C1–C3** were shown not to absorb above 280 nm, see Figure 2d) or oxazole **E** at 254 nm. Since we do not have any support for these effects and in light of what discussed above, I would prefer to keep the manuscript as it is.

3) The mechanistic rationale behind the effect of 2,6-lutidine on yield improvement remains unclear. Although the authors cite related work by Taniguchi et al., a more detailed discussion is needed to explain the observed increase in yield from 55% to 85%. It would also be helpful to explore the effect of lower equivalents (e.g., 0.1 eq), given that 0.2 eq outperformed higher loadings.

The reaction indeed proceeds in MeOH without additives, affording oxazole **4b** in 55% yield. Addition of 2,6-lutidine significantly enhances the outcome, with 0.2 equiv. giving the optimal 85% yield. When we tested 0.1 equiv., the yield decreased to 72%, whereas higher loadings were less effective. We agree that the precise mechanistic rationale remains elusive. Based on the precedent by Taniguchi *et al.* and our own observations, a plausible explanation is that 2,6-lutidine transiently buffers against acid build-up or engages in specific interactions with reactive intermediates, thereby suppressing side pathways. The non-linear dependence on the amount of additive suggests that its beneficial effect is subtle and may involve a balance between proton management and undesired coordination. Unfortunately we do not see any clear interaction between the lutidine and either the starting material or the product and we cannot detect the transient intermediates so at the moment we cannot provide any more conclusive rationale.

4) While the substrate scope is broad, discussing the method's limitations would enhance its practical relevance. Specifically, it would be valuable to explore common functional groups in drug-like molecules that may not tolerate the reaction conditions (e.g., amides, lactones) and to include examples of suboptimal or unsuccessful transformations.

I agree with the reviewer on this but I do not think the key aspect is which functional groups are tolerated, but rather how they impact the reactivity based on where they are located on the isoxazole core. There are some non-working examples in Figure 4 as the work was designed to capture this aspect. As an example, even a Me-group, which is not considered a functionality, can have a dramatic impact on the photochemistry depending on where it is placed. We do not have examples that did not work as we evaluated the scope on the basis of the guidelines that emerged from the initial screening of all substitution patterns with Me, Ph and ester substituents.

5) If feasible, applying the divergent isoxazole ring placement (Fig. 5) to more complex molecules such as substrates **50**, **51**, or **52** would provide better evidence of the method's generality and synthetic utility.

Unfortunately, compounds bearing substituents at positions C3 or C4 of the isoxazole core cannot be deconstructed to the corresponding α -ketonitrile, and various side reactions are observed depending on the conditions employed (see Figure 4c). Specifically, when a substituent is present at C3, the α -ketonitrile does not form; instead, only product **13c** is observed. Similarly, when a substituent is introduced at C4, we were unable to detect the α -ketonitrile product under any of the conditions tested. As correctly pointed out by the reviewer, this represents a limitation of our methodology: only "terminal" isoxazoles can currently undergo the deconstruction-reconstruction sequence to access diverse new heteroaromatic frameworks. Surprisingly, we were not able to obtain the corresponding α -ketonitrile in the case of **52a** and we currently do not have any clear explanation for this behaviour. I have discussed this aspect clearly stating the limitation in the revised manuscript.

6) The introduction would benefit from better contextualizing this work within the field of skeletal editing, which has seen significant recent advances. The authors should consider citing key recent reviews, such as: Sarpong and Levin, *Nat. Synth.*, 2022, 1, 352–364; Ball, *Helv. Chim. Acta*, 2023, 106, e202200182. These have been added

7) Although a minor issue, the current compound numbering system may cause confusion. It may be clearer to assign unique numbers to distinct skeletons and use letters (a, b, c, etc.) for derivatives.

I am not convinced this assignment will avoid confusions about the numbering. I believe with this numbering system, it is easier for the reader to identify which product comes from which substrate.

8) Please consider the commented PDF attached.

We modified the main text accordingly. See the modifications below:

page 3:

- "isothiazole" → isoxazole
- check "H-bond" and " π -stacking" (there are symbol letters that should not be there)
- "However, pioneering studies from Pavlik and other demonstrated its potential for application in isoxazole-to-oxazole isomerization although their conditions often led to complex reaction outcomes, low chemical yields, and mixtures of isomeric products." → However, pioneering studies from Pavlik and other

demonstrated its synthetic potential in isoxazole-to-oxazole isomerization although their conditions often led to complex reaction outcomes, low chemical yields, and mixtures of isomeric products.

page 6:

- An image has been added to Section 8 of the SI showing the spin densities of the vinylnitrene intermediates **L2-4**.

page 7:

- Regarding the synthesis of **11a**, I am not sure which reported procedure you are mentioning. On SciFinder, we can indeed find three procedures, one with permutation (4% yield, cited in the manuscript), one forming a mixture of isomers and using a tailored titanium catalyst (*Tetrahedron* **2012**, *68*, 807-812), and a last one where the starting material needs to be synthesized and almost no experimental details are given.
- Regarding the synthesis of **23a**, there is indeed a reported two-step synthesis in ACS Medicinal Chemistry Letters so we have cited it. This does not alter the conclusions of our work.

page 8:

- Add note for "E" in Figure 3.
- The yields obtained for all conditions are reported in the SI. If the reviewer believes it is necessary, we can add a note or a sentence in the main text highlighting this.
- Add numbers for oxazoles with vinyl and C(sp²)-Br

page 11:

- Figure 2 → Figure 3 for **2a** and Figure 4 for the others
- **4a** → **2a**

page 12:

- Figure 2 → Figure 3 & Figure 4

Reviewer #3: Leonori et al. present a photochemical platform for the selective conversion of isoxazoles into five-membered heteroaromatics by leveraging excited-state reactivity. The authors systematically investigated the effects of substitution sites during the isoxazole-to-oxazole transformation using DFT calculations. Concurrently, they conducted a scope analysis of differentially substituted isoxazoles. For the divergent "ring replacement" of isoxazoles, the key lies in the transient formation of common intermediate α -ketonitriles. By subjecting these intermediates to distinct reaction conditions, the authors observed divergent outcomes, enabling the construction of five heterocyclic frameworks through a one-flask operation.

My main concerns with this manuscript arise from several critical scientific considerations, which cumulatively cast doubt on its readiness for publication in Nature Communications. The concerns can be systematically categorized as follows:

1. The manuscript's core focus on the isoxazole-to-oxazole transformation, while methodologically sound, is circumscribed by critical precedents. As detailed in References 24 and 25 (Ullman, Singh et al.), this rearrangement class has established synthetic precedence, with Reference 31 recently reporting an isoxazole-derived oxazole synthesis under Hg-lamp irradiation. These prior disclosures inherently diminish the work's claimed novelty, despite the authors' contributions to substrate scope expansion and mechanistic elaboration. The study thus manifests as a synthetic and mechanistic refinement rather than an innovation meeting the elevated standards expected of Nature Communications.

We thank the Reviewer for their thorough assessment and for raising important points regarding the novelty of our work. We hope the following clarifications will be taken into consideration.

- We agree that the photochemical reactivity of isoxazoles has historical precedent. I am very sorry if the way I wrote the original manuscript gave the impression of dismissing prior contributions, especially the seminal work in Ref. 24, 25 & 27. However, despite several reports, the photochemistry of isoxazoles (as well as the one of many other heteroaromatics) has neither been really explored nor thoroughly understood and lacks clear predictive guidelines. We believe this work attempts to fill some of these gaps.
- Singh reported only two substrates, outlining the yield only for the first one which was diphenylisoxazole (see scheme below). Despite a relatively in-depth mechanistic analysis, the authors did not study the possible scope, substituent or solvent effect of this photoreactivity.

- Regarding Pavlik's work (Ref. 27), despite being a crucial contribution to the field, this study examined only five isoxazole substrates, and only two were irradiated on a preparative scale. The remaining were evaluated under short irradiation times, limiting the ability to understand their true reactivity. See the answer to **Reviewer #1** for more details.
- See the answer to **Reviewer #1** for more details about Ref. 31.
- Our central contribution also lies in framing isoxazoles as programmable, masked precursors to access directly multiple heteroarenes. This one-pot deconstruction–reconstruction aligns well with emerging needs in medicinal chemistry, particularly in late-stage diversification and bioisosteric replacement strategies (see *Nature Chem* 2018, 10, 383–394. <https://doi.org/10.1038/s41557-018-0021-z>). As illustrated by recent work from the McNally group (*Nature* 2024 631, 87–93. <https://doi.org/10.1038/s41586-024-07474-1>; ref. 9a in the manuscript), there is strong and growing interest in modular frameworks that enable rapid access to chemical diversity from common building blocks.

I hope to believe that our work goes beyond revisiting known transformations. We conducted a systematic and comprehensive exploration of >50 substrates, aiming to establish general reactivity trends and operationally simple guidelines that are relevant to modern synthetic applications. Indeed, aspects like the impact of Ph-substitution around the isoxazole ring or the impact of additional functionalities have never been demonstrated

and reveal a complex photochemical scenario that cannot be appreciated looking at the reactivity reported in Ref. 24, 25 & 27.

2. On page 4, line 13, the authors note that "the 3-phenyl analogue (4a) proved photochemically inert under all tested conditions, even upon prolonged irradiation," with DFT calculations attributing this to a higher energy barrier for aziridine formation. However, a discrepancy arises: substrates 8a, 20a, 21a, 48a, and 49a—all 3-phenyl-substituted isoxazoles—react smoothly under the protocol. This inconsistency necessitates mechanistic clarification. The authors are expected to perform additional DFT calculations to rationalize the reactivity differences between di-substituted and tri-substituted isoxazoles.

We have calculated the energy profiles for the thermal isomerization pathways of two additional isoxazoles, **20a** (disubstituted, 3-Ph-4-CO₂Me) and **49a** (trisubstituted, 3-Ph-4-CO₂Me-5-Me). These profiles are shown below. In both cases we could not locate any nitrene intermediate, with a key difference. For **20a**, the transition state for the N–O cleavage leads to a bifurcation that promotes the formation of two intermediates, the expected azirine **C20** and a new diradical intermediate, **I20**, in which the N undergoes *ipso* addition to the Ph ring. In contrast, **49a** leads to the exclusive formation of azirine **C49**. We believe that these computational results are in good agreement with the experimental observations. Specifically, for **4a** (monosubstituted 3-Ph-isoxazole,) N–O bond cleavage forms the vinylnitrene intermediate, for which the energy barrier to generate the azirine is prohibitively high and prevents any reactivity (see the discussion around Figure 2d). For **20a**, N–O cleavage forms either the azirine **C20**, which could evolve into the oxazole, or the biradical **I20**, which cannot and might lead to decomposition explaining the loss of ca. 40% of starting material. Finally, for **49a**, exclusive formation of the azirine **C49** promotes the formation of the oxazole without significant degradation, in accordance with <10% starting material loss observed experimentally. These results have been added to Section 8 of the SI.

3. Additionally, the observed substrate specificity warrants concern. As illustrated in Figure 3, minor substituent variations on the isoxazole scaffold lead to divergent reaction outcomes. Even for privileged aryl substrates bearing 5-substituted isoxazole moieties (e.g., 33a, 38a, 41a), reaction conditions required case-by-case optimization to enable the transformation. Such results undermine the synthetic utility of the reported method.

The photochemistry of heteroaromatic compounds such as isoxazoles is inherently sensitive to electronic and steric effects. Small changes in substitution patterns (e.g., replacing a Ph with a Me or CO₂Me) can dramatically alter excited-state behaviour, reaction pathways, and product distributions. This is not a shortcoming of the chemistry; it is a defining feature of photochemical reactivity. Expecting uniformly identical outcomes across diverse substrates ignores this core principle. Nevertheless, compounds rather diverse but belonging to the class of 5-aryl isoxazole display rather general reactivity with a single reaction condition (condition F).

4. The SI is well-prepared, but ¹H NMR spectra of 43b and 58d show impurities. Please re-purify and resubmit clean spectra.

We have attempted to re-purify these two samples but I am sorry we do not manage. However, the impurities are very minor peaks. Unfortunately, our prep-HPLC is out of order so we cannot address this comment any better.

Reviewer #4: I co-reviewed this manuscript with one of the reviewers who provided the listed reports. This is part of the Nature Communications initiative to facilitate training in peer review and to provide appropriate recognition for Early Career Researchers who co-review manuscripts.

We thank the reviewer for their positive evaluation of our work.

Reviewer #5: The manuscript presents a novel photochemical platform for the transformation of isoxazoles into structurally diverse heterocycles, including oxazoles, pyrazoles, pyrroles, and isothiazoles. This work provides significant value for the derivatization of five-membered nitrogen-containing heterocycles in pharmaceutical development. Although a robust methodological design, comprehensive data collection, and detailed DFT calculations are done to investigate the complex interplay between sterics, electronics, and photophysical properties, certain aspects of the mechanistic analysis and discussion require further clarification.

We thank the reviewer for their positive evaluation of our work.

(1) The proposed mechanism leading to the isoxazole-to-oxazole (A-to-E) isomerization includes the photoexcitation of isoxazole A, N–O bond homolysis, 2π -electrocyclicization, and second photoexcitation to produce the corresponding oxazoles. To better understanding the selectivity, the energies of the excited singlet state (S_1 -A) and the corresponding conical intersection (CI, S_1/S_0) points should be determined in the energy profiles.

We included an extended version of Figure 2d including additional information regarding the photochemical steps. Specifically, this revised image includes the energies of the S_1/S_0 conical intersections, the nitrile ylide intermediates that arise following excitation of the azirines, and the energy barriers that separate them from the oxazoles.

(2) According to the mechanism in Figure 2c, the conversion from C to E involves second photoexcitation and C–N bond cleavage to obtain nitrile ylide (D), and then through an electrocyclicization transition state to obtain the oxazole product. However, the second photoexcitation process and the detailed structure and energy information of the nitrile ylide (D) are missing in Figure 2d. As reported in Su's work (J. Phys. Chem. A 2015, 119, 9666–9669), it is separated processes for C–N bond cleavage (C to D) and C–O bond formation (D to E). Is TS3 a concerted process for these two steps or just C–O bond formation from nitrile ylide?

Please refer to our response to the first comment of reviewer #2. In a nutshell, the azirine \rightarrow oxazole thermal isomerization pathway (corresponding to TS3 in Figure 2d) is concerted, the photochemical pathway (now shown in the revised Figure 2d) is stepwise.

(3) The potential for intersystem crossing (ISC) from singlet to triplet along the reaction pathways should be considered, including the triplet states of isoxazole (T_1 -A), and the triplet states of other key intermediates, just in case the involvement of triplet-state mechanisms.

Regarding the reviewer's suggestion that a triplet excited state could be involved, we had already conducted some EnT catalysis screening experiments using a range of known triplet photosensitizers. These efforts, detailed below and in the revised Supporting Information, showed no productive reactivity – only full recovery of starting material. This outcome supports our choice to focus on direct excitation of the isoxazole core and a singlet excited state mechanism, which ultimately proved more effective under the current conditions.

entry	PC	solvent	wavelength (nm)	rsm (%)	2b (%)	2c (%)
1	[Ru(bpz) ₃][PF ₆] ₂	DCE	440 nm	> 95%	n.d.	n.d.
2	[MesAcr]ClO ₄	DCE	440 nm	> 95%	n.d.	n.d.
3	4CzIPN	DCE	440 nm	> 95%	n.d.	n.d.
4	[Ir{dF(CF ₃)ppy} ₂ (dtbpy)]PF ₆	DCE	390 nm	> 95%	n.d.	n.d.
5	fac-Ir(ppy) ₃	DCE	390 nm	> 95%	n.d.	n.d.
6	[Ru(bpz) ₃][PF ₆] ₂	CH ₃ CN	440 nm	> 95%	n.d.	n.d.
7	[MesAcr]ClO ₄	CH ₃ CN	440 nm	> 95%	n.d.	n.d.
8	4CzIPN	CH ₃ CN	440 nm	> 95%	n.d.	n.d.
9	[Ir{dF(CF ₃)ppy} ₂ (dtbpy)]PF ₆	CH ₃ CN	390 nm	> 95%	n.d.	n.d.
10	fac-Ir(ppy) ₃	CH ₃ CN	390 nm	> 95%	n.d.	n.d.

(4) The NMR spectral data require the following corrections: The ^1H NMR integrals need calibration, as exemplified by the unlabeled integrations for peaks 8a and 18a. Additionally, the notation ' ^{13}C NMR' for compound 52a (151 MHz, CDCl_3) should be revised to ' ^{13}C NMR.' Furthermore, the ^1H NMR integrals for peaks 39a/b must be properly adjusted for accuracy.

See above – I am sorry but we have not managed to remove these very minor impurities from the samples.

Major Revision The authors should address the above concerns, particularly the mechanistic clarifications and the explanation for the energy barrier/yield discrepancy, before the manuscript can be reconsidered for publication. The revisions are essential for ensuring the robustness and clarity of the proposed mechanism.

Reviewer #2 (Remarks to the Author):

The manuscript has improved, and I appreciate the authors' efforts. However, I still have two major concerns: (i) the synthetic value/generalizability remains uncertain, as small substituent changes seem to cause large outcome shifts and performance on complex, drug-like substrates appears limited; and (ii) the mechanistic basis of the differential reactivity of 3a and 2a leans mainly on TD-DFT absorption profiles of intermediates (not a solid evidence and doesn't explain the whole reaction profile), and does not yet offer a solid explanation for solvent/additive effects.

Response to point 1.

Our study includes 55 examples, covering mono-, di-, and tri-substituted isoxazoles, thus encompassing all types of substitution classes. We demonstrate compatibility with a wide range of substituents, including alkyl, electron-rich and electron-poor aryl, heteroaryl (both electron rich and electron poor – indole, thiophene, pyridine...), and ester substituents directly attached to the isoxazole ring. Furthermore, we show tolerance of diverse functional groups such as Br, Cl, B(pin), cyclopropane, ether, carboxylic acid, and sulfonamide. We have illustrated the synthetic relevance of this methodology through:

- The two-step synthesis of a bioactive compound that previously required an eight-step sequence.
- Successful permutation of two marketed drugs.

Photochemical sensitivity to substitution is an intrinsic property of excited-state processes and does not, per se, imply lack of generality. To address this explicitly, we systematically evaluated substitution at all three positions of the isoxazole core across 55 examples and identified well-defined structural domains where the reaction is robust and high-yielding. Once these domains were established, extended scope studies were performed within these regions of reactivity. The successful permutation of two marketed drugs and the streamlined synthesis of a bioactive target further demonstrate that the method is applicable to structurally complex, medically relevant substrates within these defined boundaries.

Response to point 2.

The key intermediates likely involved in the reactivity are transient and cannot be isolated or detected, even when probed using laser flash photolysis. TD-DFT calculations therefore represent the only viable approach. While TD-DFT alone cannot provide a complete kinetic profile, it successfully reproduces the relative absorption trends and rationalizes the distinct photochemical fates of **3a** and **2a**. Experimental observations (quenching, selectivity, and control experiments) are consistent with these theoretical insights.

Regarding solvent and additive effects, this remains a complex and open field in photochemistry. The interplay between polarity, hydrogen-bonding, and solvation of excited states is often non-linear and system-dependent. Even in well-studied systems (e.g., Pd-catalyzed C–H activation in HFIP, see "Hexafluoroisopropanol: the magical solvent for Pd-catalyzed C–H activation" *Chem. Sci.* **2021**, *12*, 15773), the mechanistic origins of solvent effects remain incompletely understood. I started a collaboration with Prof. Davide Avagliano (Chimie ParisTech) to model these phenomena in detail; however, this is a substantial computational challenge that will require dedicated future work and goes beyond the scope of the present manuscript.

Reviewer #3 (Remarks to the Author):

The authors have addressed most of my concerns, yet some aspects still require further refinement.

Below are my comments on the resubmitted manuscript.

The authors stated that “Small changes in substitution patterns (e.g., replacing a Ph with a Me or CO₂Me) can dramatically alter excited-state behaviour, reaction pathways, and product distributions.” While it is reasonable to anticipate the need for reaction condition adjustments when a phenyl group is substituted with a methyl or ester group—given the distinct electronic and steric properties of these moieties—the fact that 33a, 38a, and 41a all fall within the 5-aryl isoxazole scaffold undermines the persuasiveness of this explanation.

I am sorry but I do not understand what they mean with this comment and how these 3 compounds undermine the claim. The sentence cited by the reviewer is related to substituents directly attached to the scaffold.

In the ¹H NMR spectrum of compound 58d, both solvent peaks and impurity peaks are non-negligible, as they can significantly interfere with the accurate determination of the compound's purity—and thus indirectly affect the reliable calculation of its yield. If further purification proves unfeasible, this substrate should be excluded from subsequent studies.

The NMR spectrum of compound 58d has been re-acquired after additional purification and now meets standard purity and reporting criteria.

Additionally, after carefully reviewing the authors' responses to Reviewer #1, this reviewer confirms that the latter's technical requests have been properly addressed.

Thank you.

[Note from the Editor: Reviewer #3 was asked to assess also the response given to Reviewer #1 who was unable to look over the revision]

Reviewer #4 (Remarks to the Author):

Reviewer #5 (Remarks to the Author):

In the new version, the authors did a more detailed mechanistic study, both in calculation and experimentally. As shown in Figure 3, the second photoexcitation process and the detailed structure and energy information of the nitrile ylide are calculated. The energies of the S1/S0 conical intersections are also added to support the proposed mechanism. The triplet-state mechanisms are excluded by EnT catalysis screening experiments using a range of known triplet photosensitizers, but if the authors have calculated the energies of triplet states of the key intermediates. Are they also consistent with the experimental results?

After the author's efforts, most of my concerns have been addressed in the new revisions. The revised manuscript could be considered for publication.

Thank you.

I am very sorry not to have responded to the first point from Reviewer 5. We had calculated all values and not sure why I didn't include it... Here it is everything that supports the lack of reactivity we observed using triplet energy transfer catalysis.

Reviewer #5 (Remarks to the Author):

In the new version, the authors did a more detailed mechanistic study, both in calculation and experimentally. As shown in Figure 3, the second photoexcitation process and the detailed structure and energy information of the nitrile ylide are calculated. The energies of the S1/S0 conical intersections are also added to support the proposed mechanism. The triplet-state mechanisms are excluded by EnT catalysis screening experiments using a range of known triplet photosensitizers, but if the authors have calculated the energies of triplet states of the key intermediates. Are they also consistent with the experimental results?

We have calculated the ET for several isoxazoles:

Isoxazoles	2a (5-Ph)	3a (4-Ph)	4a (3-Ph)
E_T (kcal mol ⁻¹)	66.7	74.2	74.0

These ET values are significantly higher than the ones of the standard EnT photocatalysts commonly employed in photocatalysis.

Photocatalysts	[Ru(bpz) ₃] ²⁺	[MesAcr] ⁺	4CzIPN	[Ir{dF(CF ₃)ppy} ₂ (dtbpy)] ⁺	fac-Ir(ppy) ₃
E_T (kcal mol ⁻¹)	48.4 [1]	44.7 [2]	58.3 [1]	61.8 [1]	58.1 [1]

[1] [10.1039/D3CS00190C](https://doi.org/10.1039/D3CS00190C)

[2] [10.1021/ja052967e](https://doi.org/10.1021/ja052967e)